# Influence of Ambient Air Pollution on Rheumatoid Arthritis Disease Activity Score Index

**DOI:** 10.3390/ijerph17020416

**Published:** 2020-01-08

**Authors:** Ahmad Alsaber, Jiazhu Pan, Adeeba Al-Herz, Dhary S. Alkandary, Adeeba Al-Hurban, Parul Setiya

**Affiliations:** 1Department of Mathematics and Statistics, University of Strathclyde, Glasgow G1 1XH, UK; 2Department of Rheumatology, Al-Amiri Hospital, P.O. Box 4077, Safat 13041, Kuwait; adeebaalherz@yahoo.com; 3Department of Earth and Environmental Sciences, Kuwait University, P.O. Box 25944, Safat 1320, Kuwait; dhary.alkandary@ku.edu.kw (D.S.A.); Q8geo@hotmail.com (A.A.-H.); 4Department of Agrometerology, Govind Ballabh Pant University of Agriculture & Technology, Pantnagar, Uttarakhand 263145, India; parul.setiya@gmail.com

**Keywords:** air pollution, rheumatoid arthritis, DAS-28, CDAI, RA registry, KRRD, Kuwait, Middle East, AQI

## Abstract

Rheumatoid arthritis (RA) is a chronic autoimmune of an unknown etiology. Air pollution has been proposed as one of the possible risk factors associated with disease activity, although has not been extensively studied. In this study, we measured the relationship between exposure to air pollutants and RA activity. Data on RA patients were extracted from the Kuwait Registry for Rheumatic Diseases (KRRD). Disease activity was measured using disease activity score with 28 examined joints (DAS-28) and the Clinical Disease Activity Index (CDAI) during their hospital visits from 2013 to 2017. Air pollution was assessed using air pollution components (PM${}_{10}$, NO${}_{2}$, SO${}_{2}$, O${}_{3}$, and CO). Air pollution data were obtained from Kuwait Environmental Public Authority (K-EPA) from six different air quality-monitoring stations during the same period. Multiple imputations by the chained equations (MICE) algorithm were applied to estimate missing air pollution data. Patients data were linked with air pollution data according to date and patient governorate address. Descriptive statistics, correlation analysis, and linear regression techniques were employed using STATA software. In total, 1651 RA patients with 9875 follow-up visits were studied. We detected an increased risk of RA using DAS-28 in participants exposed to SO${}_{2}$ and NO${}_{2}$ with β=0.003 (95% CI: 0.0004–0.005, p<0.01) and β=0.003 (95% CI: 0.002–0.005, p<0.01), respectively, but not to PM${}_{10}$, O${}_{3}$, and CO concentrations. Conclusively, we observed a strong association between air pollution with RA disease activity. This study suggests air pollution as a risk factor for RA and recommends further measures to be taken by the authorities to control this health problem.

## 1. Introduction

Rheumatoid arthritis (RA) is a chronic autoimmune disease that mainly affects the joints causing inflammation, pain, and difficulty to use the joints. Although the exact cause is unknown, many genetic and environmental factors have been linked to the disease. Exposure to chemicals has previously been proposed as a possible, if not the main cause of the disease [1].

Several studies suggested that exposure of air pollution may increase the risk of RA [2,3]. Furthermore, epidemiological evidence indicates a significant association between the risk of RA and exposure to environmental factors, such as cigarette smoke, dioxin, noise, and traffic-related air pollution [4,5,6].

Pollutants with the strongest evidence for public health concern include *particulate matter* (PM), *ozone* (O${}_{3}$), *nitrogen dioxide* (NO${}_{2}$), and *sulfur dioxide* (SO${}_{2}$) [7]. However, few studies have followed adequate methodologies correlating meteorological variables with RA. In this context, further investigation concerning the impact of air pollution on the risk of developing RA is still necessary.

With respect to RA, there is a need to investigate the impact of air pollution on RA through elaborate research. Air pollutants are part of the environmental components resulting from dust storms and fossil fuel combustion, which determine RA symptoms and worsen the overall disease. Additionally, gases such as SO${}_{2}$, NO${}_{2}$, *carbon monoxide* (CO), and O${}_{3}$ are the other main pollutants that cause RA [8]. More in general, extensive investigations have been performed by several researchers about the impact of ambient air pollution on human health [9,10]. For instance, SO${}_{2}$ resulting from the combustion of fossil fuels with high sulfur content is considered one of the most common pollutants with the worst impact on air quality. The combustion of fossil fuels containing high sulfur content causes the release of sulfur dioxide into the atmosphere. Several researchers have adequately documented the harmful effects of long-term exposure to high levels of SO${}_{2}$ on overall health [11,12]; therefore, increased emission of pollutants into the atmosphere may potentially result in several adverse health effects, including RA.

Many composite indices for RA progress measurement are actually available: for example, the Disease Activity Score (DAS) with 28 examined joints (DAS-28) is one common RA index that has been extensively employed to identify the disease progress level for RA patients [13,14,15]. The Disease Activity Score was developed to measure and assess RA disease activity in daily clinical practice, clinical trials, and long-term observational studies [16]. The second RA index is the Clinical Disease Activity index (CDAI) [17,18], which is used to assess disease activity. The CDAI was developed to provide physicians and patients with simple and more understandable instruments.

In the present study, we aimed to investigate whether exposure to ambient air pollution (i.e., PM with aerodynamic diameter < 10 µm (PM${}_{10}$), NO${}_{2}$, SO${}_{2}$, O${}_{3}$, and CO) is associated with an increased risk of RA using the DAS-28 and CDAI indices.

## 2. Materials and Methods

### 2.1. Patients Data Source (Kuwait Registry for Rheumatic Diseases—KRRD)

The State of Kuwait is a small country with a total area equal to 17,818 km${}^{2}$ located on the far west side of the Asian continent. The total population of Kuwait is around 4.6 million, distributed into six main governorates [19]. All RA patients represented in this study were officially registered from the Kuwait Registry for Rheumatic Diseases (KRRD) from 2013 until the end of 2017. The KRRD is a national registry listing adult patients with rheumatic diseases. Patients who fulfilled the American College of Rheumatology (ACR) criteria for RA [20] registered from January 2013 to December 2017 were included in the study. The RA information data were collected from the rheumatology departments of four major government hospitals in Kuwait based on patient visits. The selected hospitals are mainly distributed in different governorates covering the ethnic diversity of the Kuwaiti population. The KRRD, from which this study originated, was approved by the Ethics Committees of the Faculty of Medicine at Kuwait University, and the Ministry of Health. Additionally, official consent was obtained from all represented patients enrolled in the registry [21].

### 2.2. Calculating RA Indices

RA disease activity scores are measured using two different indices: DAS-28 and CDAI. The DAS-28 is the sum of four outcome parameters: TJC28, the number of tender joints (0–28); SJC28, the number of swollen joints (0–28); ESR, the erythrocyte sedimentation rate (in mm/h) (C-reactive protein (CRP) may be used as an alternative to ESR in the calculation); and GH, the patient global health assessment (from 0 = best to 100 = worst) (see Equation (Equation 1)).

The second index is The Clinical Disease Activity Index (CDAI). CDAI takes into account the following items: TJC28, the number of tender joints (0–28); SJC28, the number of swollen joints (0–28); PaGH, the patient global health assessment (from 0 = best to 10 = worst); and PrGH, the care provider global health assessment (from 0 = best to 10 = worst) (see Equation (Equation 2)).
(1)$$\mathrm{DAS}-28=0.56\times \sqrt{TJC28}+0.28\times \sqrt{SJC28}+0.70\times \ln\left(ESR\;Or\;CRP\right)+0.014\times GH$$
(2)CDAI=TJC28+SJC28+PaGH+PrGH

### 2.3. Ambient Air Pollutants Data (Environmental Public Authority of Kuwait—K-EPA)

Pollutants data (PM${}_{10}$, NO${}_{2}$, SO${}_{2}$, O${}_{3}$, and CO) were obtained from six fixed monitoring stations run by the Environmental Public Authority of Kuwait (K-EPA). The air pollutant measurement sampling was from 1 January 2013 to 31 December 2017 based on hourly observations.

The pollutant data were distributed throughout the residential areas where stations were measuring different parameters including PM${}_{10}$, NO${}_{2}$, SO${}_{2}$, O${}_{3}$, and CO. The average concentration time of PM${}_{10}$, NO${}_{2}$, and SO${}_{2}$ was twenty-four hours, and eight-hour time-average concentrations of O${}_{3}$ and CO were collected for each measurement place, relying on 75% of the contributing values being present as the calculated averages.

According to Johnson et al. [22], the Air Quality Index (AQI) is defined as a measure of the condition of air relative to the requirements of one or more biotic species or to any human need [22,23]. The AQI is divided into categories, in which they are numbered, and each slot is marked with a color code. This provides a scale from a healthy level of zero to a very hazardous level of above 300 as a health risk indicator associated with air quality.

In this study, the air quality was assessed using the AQI developed by Al-Shayji et al. [24] for the State of Kuwait, based on the guidelines proposed by the United States Environmental Protection Agency (USEPA) [25]. The AQI is an index for reporting the day-to-day air quality. It gives details about the cleanliness of ambient air. The following equation was used to convert from pollutant concentration to AQI:(3)$$I_{p}=\frac{I_{high}-I_{low}}{C_{high}-C_{low}}\left(C_{p}-C_{low}\right)+I_{low},$$
where $I_{p}$ is the Air Quality Index for the given pollutant, $C_{p}$ is the pollutant concentration, $C_{low}$ is the concentration breakpoint that is $\leq C_{p}$, $C_{high}$ is the concentration breakpoint that is $\geq C_{p}$, $I_{low}$ is the index breakpoint corresponding to $C_{low}$, and $I_{high}$ is the index breakpoint corresponding to $C_{high}$ [26] (see Table 1).

### 2.4. Air Pollution Data Processing and Treatment

AQI data were examined by checking the normality assumption and detecting for any possible outliers before any statistical analysis or testing between the variables were done. About 5.8%, 1.6%, 48.4%, 5.3%, and 6.5% of data for PM${}_{10}$, NO${}_{2}$, SO${}_{2}$, O${}_{3}$, and CO were missing, respectively. Multiple imputations were performed to improve the accuracy of AQI prediction, where a final estimate was composed of the outputs of several multivariate fill-in methods [27,28].

The imputation process was determined by considering log transformation for better prediction to estimate and filling the missing data for each pollutant. Finally, AQI with RA information were matched with patients’ hospital location and date of patient visit. All AQI data for every pollutant were aggregated from hourly to daily observation.

### 2.5. Matching Procedure between Patients and AQI

All data management and combination was conducted using R Studio Version 1.1.463 running R 3.5.1 GUI 1.70 [29] software. Various R packages were used to clean, match, and combine the two datasets, including plyr [30], dplyr [31], tidyr [32], and stringr [33].

The matching procedure was done using a developed R code to match between RA patient information and AQI monitoring station using date and governorate variables for both KRRD and K-EPA. As mentioned above, air pollution information was taken from six different stations distributed into all six governorates in the state of Kuwait. The matching procedure was conditioned on the date of patient visit with the date of the daily average AQI using a developed R code grouped by governorate physical address (e.g., if a patient lived in the Ahmadi governorate, the AQI information that came from the Ahmadi monitoring station was added to their visit information after matching the same date; see Figure 1).

### 2.6. Statistical Analysis

In the current study, means, standard deviations (SDs), and percentages were used to summarize and compare RA characteristics between the governorate levels. To estimate the association between the pollutants and RA indices, hierarchal linear model (HLM) analysis was performed using a regression approach. Pearson correlation test was performed to highlight the significant associations between AQI for NO${}_{2}$, CO, PM${}_{10}$, SO${}_{2}$, and O${}_{3}$ and with the RA indices (DAS-28 and CDAI). Regression analyses were performed separately for DAS-28 and CDAI as response variables. Four regression models were estimated to highlight the most significant variables associated with the response variables (DAS-28 or CDAI). Because the variables belong to two different databases, we started with the first regression (M1) that measured the association between patients demographics (e.g., gender, RA disease duration, nationality, governorate, and comorbidity) with the response variables (DAS-28 or CDAI), and then we estimated the second regression model (M2) that measured the association between rheumatoid factors after adjusting for gender, RA disease duration, nationality, governorate, and comorbidity with the response variables (DAS-28 or CDAI). Models 1 and 2 were estimated from KRRD database. Then, we estimated Models 3 and 4 to highlight if there is any association between air pollution with disease activity indices (DAS-28 or CDAI) after merging the EPA with KRRD databases. Model 3 (M3) measured the direct effect from air pollutants to disease activity indices (DAS-28 or CDAI), whereas Model 4 (M4) measured the association between air pollutants to disease activity indices after adjusting for the rheumatoid factors mentioned in Model 2 (M2) (e.g., comorbidity, treatment class, swollen, tender, etc.). For better data fit, model comparison techniques using deviance scores were implemented to confirm the best choice of model [34,35]. All statistical procedures were performed using Stata 15.1 SE version software (StataCorp, College Station, TX, USA).

Model 1 (M1) was made to explain the influence of demographic variables (Disease Duration, Gender, Governorate, Nationality, Comorbidity, and Treatment Class) on the response variables (DAS-28 and CDAI). Model 2 (M2) was made to determine the effect of RA factors (Swollen, Tender, RF (rheumatoid factor), ACPA, Patient Global Assessment, and Physician Global Assessment) plus demographics on the response variables. Model 3 (M3) was made to estimate the relationship between the increase of NO${}_{2}$, CO, PM${}_{10}$, SO${}_{2}$, and O${}_{3}$ concentrations and the response variables (DAS-28 and CDAI). Model 4 (M4) was made to explain the effect of AQI in terms of NO${}_{2}$, CO, PM${}_{10}$, SO${}_{2}$, and O${}_{3}$ with RA factors (Swollen, RF, ACPA (anti-cyclic citrullinated peptide), ESR, and CRP) to indicate RA disease activity.

## 3. Results

The data of RA patients visits were obtained from KRRD, and the air pollution data were obtained from K-EPA. The analysis was performed during the period from 2013 to the end of 2017. There were 1651 RA patients with 9875 follow-up visits and 13,152 daily air pollution records. Because of the matching process to combine the data from the air pollution dataset with the RA patient visits from the KRRD dataset, the final dataset had to meet the matching conditions (matching based on date and governorate) with a total of 9875 records.

Table 2 shows some information about RA patient characteristics group by governorate location in the state of Kuwait. From the results, most patients were from Fawaniya governorate (n = 4378 visits; 44.3%). Most of the patients belonged to the local country with Kuwaiti nationality (n = 5783 visits; 58.6%). Females accounted for the majority of total visits (n = 6008; 60.8%). The average RA disease duration for all patients was 9.82 years with a SD of 6.48 years.

Most of the patients were positive for rheumatoid factor (RF; n = 6881; 74.6%) and positive for anti-cyclic citrullinated peptide (ACPA; n = 4934; 60.5%). The majority of RA patients presented co-morbidities (e.g., hypertension, hyperlipidemia, diabetes mellitus, chronic kidney disease, coronary artery disease, cancer, or any other illness; n = 5393; 54.6%). From the results, most of the consumed drugs were biologics (n = 5214; 52.8%).

In Table 2, we can see that most RA patients visited Al-Amiri hospital, with total visits n = 5051 (51.1%). Al-Amiri hospital is the only hospital in the Kuwait City governorate. The second-most patients were from Farwaniya hospital, with total visits numbering n = 3981 (40.3%). Farwaniya Hospital is the only public general hospital in Al Farwaniyah governorate. In Kuwait, there are six governorates, and each governorate has only one hospital.

With regard to disease activity, results in Table 2 present the clinical features of all RA patient visits. The average DAS-28 score was 2.67 with SD 1.26, and the average score of CDAI was 6.24 with a SD of 9.96, both indicating a low disease activity. The average and SD scores for ESR and CRP were $\bar{x}=27.19$ mm/h and SD =21.79 and $\bar{x}=6.32$ mg/L and SD =4.85, respectively, which were both within the normal ranges according to our laboratory. Moreover, the average and SD for swollen and tender joints for all RA patient visits were $\bar{x}=0.69$ and SD =2.26 and $\bar{x}=2.87$ and SD =5.60, respectively.

Table 3 shows the average air pollutant concentrations using AQI scores. The mean and SD AQIs for PM${}_{10}$, CO, NO${}_{2}$, O${}_{3}$, and SO${}_{2}$ were 167.62±214.27, 1.28±0.61, 47.98±26.64, 17.76±8.94, and 15.87±17.66, respectively. The mean exposure levels for PM${}_{10}$, CO, NO${}_{2}$, O${}_{3}$, and SO${}_{2}$ in Group 2 were 158.51±68.02, 1.31±0.44, 42.74±18.83, 18.17±7.56, and 13.94±12.04), respectively. Figure 2 shows the monthly AQI average time series for PM${}_{10}$, CO, NO${}_{2}$, O${}_{3}$, and SO${}_{2}$.

Moreover, all pollutants distribution are positively skewed (i.e., the means are higher than the medians for all pollutants). However, log transformation was employed to all pollutants in the regression model for better and quality parameter estimation. Logarithmic transformation pulls extreme values in the pollutants into the normal distribution. In the regression modeling, the logarithmic transformation is not considered, as linear explanation of the regression model is made easier to transform all other variables in the model with the same logarithmic transformation. However, this is not necessary as long as all other variables achieved the normality assumption.

Correlation analysis using Pearson correlation test was conducted to highlight the significant relationship between the pollutants (PM${}_{10}$, NO${}_{2}$, SO${}_{2}$, O${}_{3}$, and CO). Table 4 shows significant positive correlations between NO${}_{2}$ and CO ($r_{p}$ = 0.22), NO${}_{2}$ and SO${}_{2}$ ($r_{p}$ = 0.51), and O${}_{3}$ and PM${}_{10}$ ($r_{p}$ = 0.08). Significant negative correlations were discovered between O${}_{3}$ and NO${}_{2}$ ($r_{p}$ = −0.12), PM${}_{10}$ and NO${}_{2}$ ($r_{p}$ = −0.12), PM${}_{10}$ and SO${}_{2}$ ($r_{p}$ = −0.03), and PM${}_{10}$ and CO ($r_{p}$ = −0.05).

Table 4 presents the Pearson correlation coefficients between different air pollutants and RA variables. For the score of RA disease activity using the DAS-28 index, the correlation results showed a positive significant correlation with exposure of SO${}_{2}$ using AQI ($r_{p}=0.07$), and the same results were returned with the with the exposure to NO${}_{2}$ using AQI ($r_{p}=0.07$). As for particular pollutants, only Hart et al. [36] provided evidence of elevated risks for NO${}_{2}$ and SO${}_{2}$, especially in terms of seronegative RA. Other pollutants (PM${}_{10}$, CO, and O${}_{3}$) did not show any significant correlation. For the CDAI, the correlation results showed a positive significant correlation with exposure to SO${}_{2}$ using AQI ($r_{p}=0.10$), and the same results were returned with the exposure to NO${}_{2}$ ($r_{p}=0.11$). Other pollutants (PM${}_{10}$, CO, and O${}_{3}$) did not show any significant correlation with CDAI. For PM${}_{10}$, Hart et al. [36] reported an elevated odds ratio (OR), which failed to reach statistical significance. The effects of both $PM_{2.5}$ and PM${}_{10}$ were non-significant across the analyses, but were consistently more pronounced for seronegative RA.

Table 5 and Table 6 present the hierarchal linear model (HLM) analysis using multiple linear regression. The first model (M1) demonstrates the effect of patient demographics on RA disease activity, where DAS-28 and CDAI were the response variables (see Table 5 and Table 6). For both DAS-28 and CDAI, the demographics explained the disease activity scores ($R_{DAS-28}^{2}=0.034$ and $R_{CDAI}^{2}=0.030$). RA Disease Duration was not significant for DAS-28, but it had a significant association with CDAI (see Table 6 (M1)). Model 2 (M2) shows that RA factors (Swollen, Tender, RF, ACPA, Patient Global Assessment, Physician Global Assessment, ESR, and CRP) plus demographics affected DAS-28 ($R^{2}=0.865$) and CDAI ($R^{2}=0.924$). In the CDAI model (M2), Gender and RA Disease Duration were not significant in explaining the CDAI. Model 3 (M3) demonstrated and highlighted the effects of gaseous pollutants (PM${}_{10}$, NO${}_{2}$, SO${}_{2}$, O${}_{3}$, and CO) using AQI on RA disease activity; only SO${}_{2}$ and NO${}_{2}$ were significant risk factors for RA patients using the information of DAS-28 ($R^{2}=0.007$) and CDAI ($R^{2}=0.015$). The final model demonstrated the effect of gaseous air pollutants with RA factors (Swollen, RF, ACPA, ESR, and CRP) on RA disease activity. The AQI of NO${}_{2}$ and SO${}_{2}$ still showed positive associations with disease activity performance of RA. The positive effects of NO${}_{2}$ in Model 4 (M4) were β=0.003 (95% CI: 0.002–0.005) and β=0.048 (95% CI: 0.030–0.066) for DAS-28 and CDAI, respectively (e.g., for a 1 $\mu g/m^{3}$ increase in daily concentration of NO${}_{2}$, DAS-28 index will increase by 0.002 (95%CI:0.002–0.005) and CDAI index will increase by 0.048 (95% CI: 0.030–0.066)), whereas, for SO${}_{2}$, the results showed a positive significant effect with β=0.003 (95% CI: 0.0004–0.005) and β=0.044 (95% CI: 0.018–0.070) for DAS-28 and CDAI, respectively (e.g., for 1 $\mu g/m^{3}$ increase in daily concentration of SO${}_{2}$, DAS-28 index will increase by 0.003 (95%CI: 0.0004–0.005) and CDAI index will increase by 0.044 (95% CI: 0.018–0.070)).

## 4. Discussion

Air pollution is a major concern for human health, since it is known to trigger and/or induce several pathologies and subsequently to increase morbidity and mortality rates, particularly in the Middle Eastern countries such as Kuwait. Therefore, air pollution control is a crucial element that should be prioritized by governments. According to the World Health Organization (WHO), six major air pollutants including NO${}_{2}$, CO, PM${}_{10}$, SO${}_{2}$, O${}_{3}$, and lead (Pb) were identified to be primary and secondary pollutants with health risks. Many studies have found connections between particulates in the air and rates of hospitalization, chronic obstructive pulmonary disease, and restricted activity due to illness [7].

RA is considered as the most common chronic systemic auto-immune disease affecting joints, musculoskeletal apparatus, and fibrous tissues [37], whose incidence is expected to follow a positive trend during the following years [38].

Our findings show a significant relationship between the concentration of certain air pollutants and RA disease activity: particularly, our results show that increased SO${}_{2}$ and NO${}_{2}$ concentrations may increase the risk of developing RA disease activity using the index DAS-28 or CDAI. These results agree with those reported in other studies [36,39].

Concerning PM${}_{10}$, O${}_{3}$, and CO, our results do not show any effect being significant enough for considering them as risk factors of RA disease activity in terms of DAS-28 index; however, O${}_{3}$ was found to be a risk factor together with SO${}_{2}$ and NO${}_{2}$ in terms of predicting RA disease activity using CDAI. A study from Korea reported an increased risk for developing RA in adults exposed to increased O${}_{3}$ and CO concentrations [40].

For better understanding, the major sources of air pollutants in Kuwait are oil refineries, traffic, and power plants (mostly using fossil fuels), which are thought to be the main sources of SO${}_{2}$ and NO${}_{2}$ in Kuwait City [41]. Indeed, urban pollution results from various sources, including industrial activities and traffic within cities: more specifically, the most prevalent pollutants generated from road transport are NO${}_{2}$, CO, SO${}_{2}$, volatile organic compounds (VOCs), and PM [42].

Particularly, Kuwait—with a population of over four million people and a fleet of more than two million vehicles, both growing rapidly—is experiencing increases in traffic volumes, trip frequency, and trip length [43]; therefore, the poor air quality outdoors is becoming a major concern, especially for people living in Kuwait City. Kuwait is a relatively small country, with a high quantity of fixed and mobile sources of different pollutants affecting air quality, especially with a high level of road traffic. As a major oil supplier in the region, fossil fuel combustion products emitted by power stations and water distillation plants are liable for the elevated atmospheric SO${}_{2}$ concentrations, whereas VOCs and NO${}_{2}$ concentrations are mainly increased by automobiles [41].

In the present study, the relation between air pollutants and RA disease activity was measured using several regression models, to ensure that this association was still present even after the addition of RA factors that were highly significant for DAS-28 and CDAI. From Model 1 for both DAS-28 and CDAI, demographics controlled the disease activity level; these results comply with patient-reported outcomes used by the National Databank for Rheumatic Diseases (NDB) [44,45].

The results of Model 1 did not show any significant evidence concerning the effect of disease duration on the disease activity level as measured by the DAS-28; this also agrees with another study [46]. Nonetheless, the results of Models 3 and 4 confirmed the existence of a significant association between exposure to SO${}_{2}$ and NO${}_{2}$ and increase of RA disease activity: more specifically, in Model 3, where the effect of air pollutants was presented without adding RA variables, SO${}_{2}$ and NO${}_{2}$ showed significant relationships with RA’s disease activity; in Model 4, where other RA factors were included, SO${}_{2}$ and NO${}_{2}$ remained risk factors for RA disease activity level.

The Swedish Epidemiological Investigation examined the impact of prolonged exposure to air pollution on the probability of having RA: from this research, there was no evidence of a higher risk of RA resulting from exposure to PM; additionally, the overall risk of RA mildly increased due to atmospheric pollutants. This research confirmed the presence of a higher risk of RA following increases in NO${}_{2}$ and SO${}_{2}$ concentrations in ambient air [36].

Among these various fields of research, a study using distance-to-road as a proxy component for exposure to traffic-associated pollution reported a substantial increase of RA risk in those residing within shorter distances from the road [4]. Moreover, a Canadian nested case-control study reported an augmented risk of developing RA following exposure to O${}_{3}$ [5]. In another case-control study, a significant increase in the risk of RA was reported in line with increasing concentrations of NO${}_{2}$ and SO${}_{2}$ [36]. Furthermore, other studies suggested an increased risk of RA in subjects who were exposed to NO${}_{2}$, particularly in women [1], as well as positive associations between O${}_{3}$, CO, and NO${}_{2}$ and RA incidence [47].

This study has some strengths such as combining patients records with air pollution concentration to initiate a complete dataset that could be used for future academic studies. In addition, dealing with missing values using MICE algorithm increased the accuracy of the estimated regression coefficients. This study used KRRD database that includes data on RA patients in Kuwait with all previous records. One of the Limitations in the study is related to the records of the follow-up visits. As the data were extracted from a registry, the number of hospital visits is not equal for all patients. It ranges from 1 to 49 visits. Because of this limitation, time series analysis during patients visits could not be performed to estimate lag effect between air pollution and RA disease activity. Single lag day effect or moving average of several previous days lag effect could not be investigated in this study because of the data layout and the study duration is very short (from 2013 to 2017). However, it could be developed by improving time series features in the future.

## 5. Conclusions

Air pollution was significantly associated with disease activity scores in RA patients. NO${}_{2}$ and SO${}_{2}$ were found to be significant risk factors for RA activity, with 7% positive correlation with disease activity indices DAS-28 and CDAI. Future research could also be based on time-series analysis by employing univariate or multivariate time series analysis. It is also recommended that researchers classify the data on air pollution and disease activity score using a cluster technique and perform an adequate cluster analysis on the data.

## 6. Ethical Approval

The authors obtained ethical approval from the Ethics Committees of The Faculty of Medicine, University of Kuwait; The Ministry of Health in Kuwait; and the Kuwait Institute for Medical Specialization (KIMS).

## Figures and Tables

**Figure 1 ijerph-17-00416-f001:**
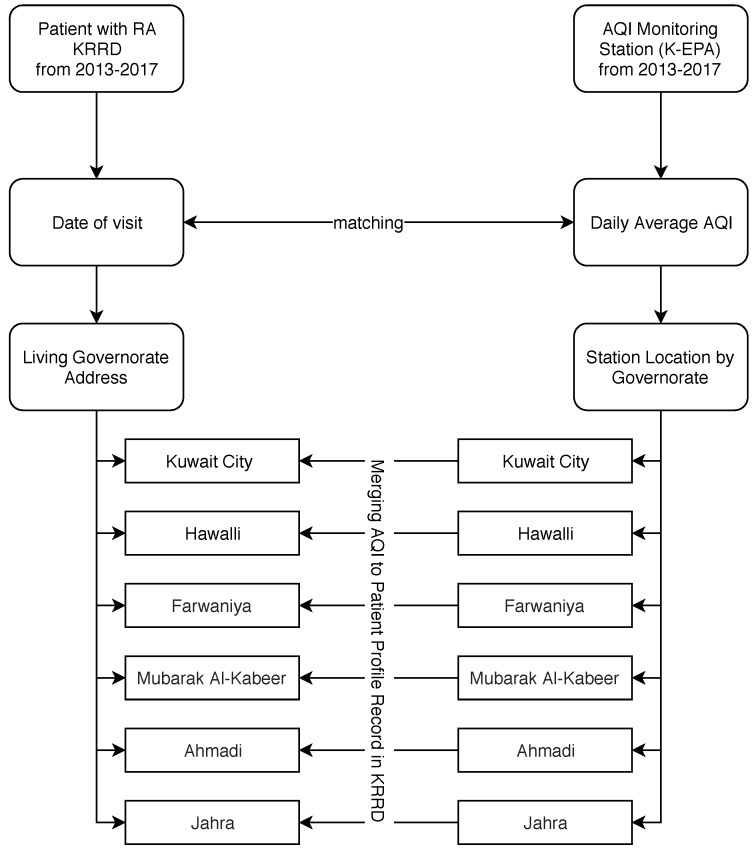
Matching Procedures to combine Air Quality Index (AQI) information with Kuwait Registry for Rheumatic Diseases (KRRD) patient profile records using date and governorate address information). K-EAPL, Environmental Public Authority of Kuwait; RA, rheumatoid arthritis.

**Figure 2 ijerph-17-00416-f002:**
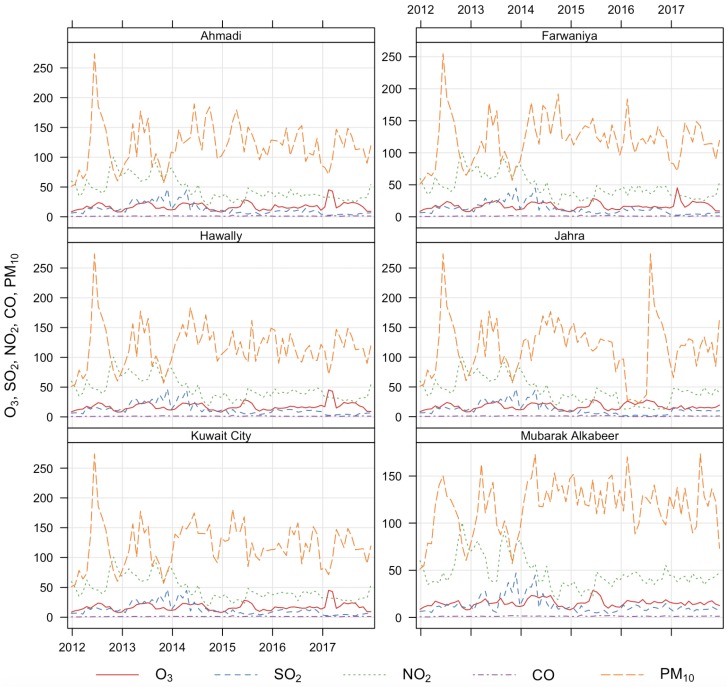
Air quality index (AQI) ambients for six governorate monitoring stations in Kuwait from 2013 to 2017.

**Table 1 ijerph-17-00416-t001:** Kuwait Air Quality Index. PM${}_{10}$: particulate matter with aerodynamic diameter ≤10 μm.

Categories	AQI, Sub-Index	O${}_{3}$ (ppm), 8-h	PM${}_{10}$ (µg/m${}^{3}$), 24-h	CO (ppm), 24-h	SO${}_{2}$ (ppm), 24-h	NO${}_{2}$ (ppm), 24-h
	$I_{\mathrm{low}}$ – $I_{\mathrm{high}}$	$I_{\mathrm{low}}$–$I_{\mathrm{high}}$	$I_{\mathrm{low}}$–$I_{\mathrm{high}}$	$I_{\mathrm{low}}$–$I_{\mathrm{high}}$	$I_{\mathrm{low}}$–$I_{\mathrm{high}}$	$I_{\mathrm{low}}$–$I_{\mathrm{high}}$
Good	0–50	0.0–0.03	0.0–90	0.0–4.0	0.0–0.03	0.0–0.03
Moderate	51–100	0.031–0.06	90.1–350.0	4.1–8.0	0.031–0.06	0.04–0.05
Unhealthy (1)	101–150	0.061–0.092	350.1–431.1	8.1–11.7	0.061–0.182	0.06–0.30
Unhealthy (2)	151–200	0.093–0.124	431.4–512.5	11.8–15.4	0.183–0.304	0.31–0.55
Very Unhealthy	201–300	0.125–0.374	512.6–675.0	15.5–30.4	0.305–0.604	0.56–1.04
Hazardous	301–500	0.375–0.504	675.1–1000	30.5–50.4	0.605–1.004	1.05–2.04

**Table 2 ijerph-17-00416-t002:** RA patient visit demographic and clinical features groups by governorate—2012 to 2017.

	[ALL]	Ahmadi	Farwaniya	Hawally	Jahra	Kuwait City	Mubarak
	**n = 9875**	**n = 356**	**n = 4378**	**n = 1272**	**n = 226**	**n = 3007**	**Alkabeer n = 636**
Gender:							
male	3867 (39.2%)	73 (20.5%)	2656 (60.7%)	285 (22.4%)	82 (36.3%)	653 (21.7%)	118 (18.6%)
female	6008 (60.8%)	283 (79.5%)	1722 (39.3%)	987 (77.6%)	144 (63.7%)	2354 (78.3%)	518 (81.4%)
Nationality:							
Kuwaitis	5783 (58.6%)	328 (92.1%)	1499 (34.2%)	773 (60.8%)	145 (64.2%)	2462 (81.9%)	576 (90.6%)
non-Kuwaitis	4092 (41.4%)	28 (7.87%)	2879 (65.8%)	499 (39.2%)	81 (35.8%)	545 (18.1%)	60 (9.43%)
Visited Hospital:							
Amiri	5051 (51.1%)	294 (82.6%)	364 (8.31%)	774 (60.8%)	120 (53.1%)	2955 (98.3%)	544 (85.5%)
Farwaniya	3981 (40.3%)	0 (0.00%)	3976 (90.8%)	5 (0.39%)	0 (0.00%)	0 (0.00%)	0 (0.00%)
Jahra	111 (1.12%)	0 (0.00%)	2 (0.05%)	3 (0.24%)	106 (46.9%)	0 (0.00%)	0 (0.00%)
Mubarak	732 (7.41%)	62 (17.4%)	36 (0.82%)	490 (38.5%)	0 (0.00%)	52 (1.73%)	92 (14.5%)
Disease Duration ${}^{3}$	9.82 (6.48)	13.3 (9.11)	9.39 (5.79)	10.4 (7.03)	10.2 (6.45)	9.42 (5.78)	11.4 (9.60)
Comorbidity ${}^{2}$:							
Yes	5393 (54.6%)	226 (63.5%)	1658 (37.9%)	802 (63.1%)	123 (54.4%)	2157 (71.7%)	427 (67.1%)
No	4482 (45.4%)	130 (36.5%)	2720 (62.1%)	470 (36.9%)	103 (45.6%)	850 (28.3%)	209 (32.9%)
Treatment Class:							
Biologics	5214 (52.8%)	327 (91.9%)	1342 (30.7%)	762 (59.9%)	131 (58.0%)	2124 (70.6%)	528 (83.0%)
cDMARDs ${}^{4}$	4661 (47.2%)	29 (8.15%)	3036 (69.3%)	510 (40.1%)	95 (42.0%)	883 (29.4%)	108 (17.0%)
RF ${}^{1}$:							
Positive	6881 (74.6%)	235 (72.8%)	3148 (76.3%)	817 (71.0%)	177 (93.2%)	2042 (72.0%)	462 (76.5%)
Negative	2348 (25.4%)	88 (27.2%)	976 (23.7%)	334 (29.0%)	13 (6.84%)	795 (28.0%)	142 (23.5%)
ACPA:							
Positive	4934 (60.5%)	102 (33.6%)	2665 (70.3%)	593 (63.1%)	73 (61.3%)	1205 (48.1%)	296 (60.0%)
Negative	3216 (39.5%)	202 (66.4%)	1125 (29.7%)	347 (36.9%)	46 (38.7%)	1299 (51.9%)	197 (40.0%)
Patient GA	1.64 (2.36)	1.54 (2.34)	1.02 (1.85)	2.60 (2.69)	2.09 (2.52)	1.95 (2.56)	2.39 (2.69)
Physician GA	1.05 (1.77)	1.06 (1.82)	0.72 (1.50)	1.63 (2.02)	1.58 (2.21)	1.13 (1.78)	1.64 (2.18)
DAS-28	2.67 (1.26)	1.85 (1.35)	2.70 (1.21)	2.77 (1.29)	3.04 (1.39)	2.61 (1.22)	2.79 (1.44)
CDAI	6.24 (9.96)	4.64 (8.89)	4.83 (8.56)	8.31(10.72)	9.45 (14.25)	6.78 (10.49)	9.00 (11.53)
ESR	27.19 (21.79)	15.12 (16.82)	30.24 (23.10)	26.77 (20.30)	30.06 (19.33)	23.97 (19.54)	27.80 (24.04)
CRP	6.32 (4.85)	4.32 (3.91)	7.29 (4.89)	4.53 (4.53)	6.36 (4.42)	6.08 (4.68)	5.38 (4.87)
Swollen Joints	0.69 (2.26)	0.34 (1.57)	1.08 (2.60)	0.53 (1.97)	0.95 (3.63)	0.26 (1.67)	0.57 (1.99)
Tender Joints	2.87 (5.60)	1.72 (4.32)	2.02 (4.18)	3.55 (6.21)	4.82 (8.39)	3.46 (6.36)	4.54 (7.13)

${}^{1}$ RF, rheumatoid factor; ACPA, anti-cyclic citrullinated peptide antibody. ${}^{2}$ Comorbidity (e.g., hypertension, hyperlipidemia, diabetes mellitus, etc.); ${}^{3}$ Disease Duration, RA disease duration by years. ${}^{4}$ DMARDs, conventional disease modifying anti-rheumatic drugs. GA, global assessment.

**Table 3 ijerph-17-00416-t003:** Distribution of Kuwait ambient air pollution exposure using AQI during 2012–2017.

Air Pollutant	Ahmadi (n = 356)	Farwaniya (n = 4378)	Hawally (n = 1272)	Jahra (n = 226)	Kuwait City (n = 3007)	Mubarak Alkabeer (n = 636)	ALL (n = 9875)
**PM** ${}_{10}$							
min	17.108	20.346	21.948	18.837	12.138	5.421	5.421
25th ${}^{1}$	75.839	76.312	80.692	67.417	71.114	74.886	74.965
median	112.623	120.779	113.586	92.788	116.081	109.917	113.586
75th ${}^{2}$	180.403	186.145	180.581	151.762	193.121	184.282	186.568
max	549.419	511.826	588.494	545.789	577.706	585.077	588.494
mean (SD ${}^{3}$)	142.36 ± 99.58	146.47 ± 94.20	145.69 ± 99.23	123.64 ± 95.00	146.81 ± 102.42	142.40 ± 102.79	144.87 ± 100.64
**CO**							
min	0.240	0.207	0.087	0.042	0.087	0.292	0.042
25th	0.938	0.945	0.984	0.854	1.006	0.978	0.975
median	1.367	1.338	1.337	1.151	1.369	1.380	1.346
75th	1.679	1.603	1.672	1.455	1.679	1.728	1.672
max	4.894	4.701	4.471	5.122	8.143	5.287	8.143
mean (SD)	1.37 ± 0.62	1.33 ± 0.57	1.40 ± 0.66	1.14 ± 0.64	1.40 ± 0.66	1.46 ± 0.70	1.39 ± 0.66
**NO** ${}_{2}$							
min	9.080	9.080	5.346	8.229	5.346	5.346	5.346
25th	25.768	28.200	27.208	29.127	27.319	32.037	27.391
median	35.762	36.537	35.810	52.420	36.795	44.744	37.355
75th	54.218	53.150	51.409	67.866	52.054	68.210	54.552
max	137.960	135.878	134.123	107.279	208.670	207.557	208.670
mean (SD)	42.85 ± 24.13	42.01 ± 20.70	41.29 ± 20.60	50.92 ± 25.54	43.00 ± 22.76	52.09 ± 27.91	43.74 ± 23.13
**O** ${}_{3}$							
min	4.051	4.051	4.051	5.887	3.476	4.877	3.476
25th	10.965	11.030	10.851	12.457	10.581	11.328	10.851
median	15.215	15.372	15.104	18.192	14.709	14.164	14.985
75th	20.874	22.240	20.451	25.172	20.451	19.507	20.759
max	54.262	69.682	80.656	37.539	88.623	57.330	88.623
mean (SD)	17.04 ± 8.75	18.53 ± 11.27	17.58 ± 10.52	18.71 ± 7.24	16.85 ± 10.08	16.26 ± 7.51	17.19 ± 9.90
**SO** ${}_{2}$							
min	0.003	1.000	1.000	0.665	0.003	1.000	0.003
25th	4.208	5.293	4.875	5.435	4.490	7.333	4.875
median	8.333	8.000	8.594	13.792	7.993	14.083	8.727
75th	14.146	17.292	17.169	24.583	16.746	22.946	17.504
max	121.833	121.833	121.833	76.875	111.917	127.875	127.875
mean (SD)	13.15 ± 15.46	13.26 ± 14.22	14.18 ± 16.10	17.90 ± 16.52	13.39 ± 14.60	18.62 ± 17.07	14.26 ± 15.42

^1^ 25th, lower quartile (25th percentile). ^2^ 75th, upper quartile (75th percentile). ^3^ SD: standard deviation.

**Table 4 ijerph-17-00416-t004:** Correlation analysis between rheumatoid arthritis disease factors and AQI for SO${}_{2}$, NO${}_{2}$, CO, O${}_{3}$, and PM${}_{10}$.

	DAS-28	CDAI	NO${}_{2}$	O${}_{3}$	SO${}_{2}$	CO	PM${}_{10}$	Swollen	Tender	ESR
DAS-28										
CDAI	0.77 ****									
NO${}_{2}$	0.07 ****	0.11 ****								
O${}_{3}$	0.00	0.00	−0.12 ****							
SO${}_{2}$	0.07 ****	0.10 ****	0.51 ****	−0.09 ****						
CO	−0.01	0.02	0.22 ****	0.02	0.07 ****					
PM${}_{10}$	0.00	−0.02	−0.12 ****	0.08 ****	−0.03 *	−0.05 **				
Swollen	0.50 ****	0.60 ****	0.01	0.01	0.01	0.00	−0.02			
Tender	0.72 ****	0.93 ****	0.13 ****	0.01	0.11 ****	0.03	−0.01	0.42 ****		
ESR	0.65 ****	0.20 ****	0.00	−0.02	0.04 *	−0.04 *	0.02	0.16 ****	0.17 ****	
CRP	0.28 ****	0.02 *	0.01	0.02	0.01	−0.01	−0.01	0.11 ****	0.02 *	0.37 ****

Note: * *p* < 0.1; ** *p* < 0.05; *** *p* < 0.01; **** *p* < 0.001.

**Table 5 ijerph-17-00416-t005:** Coefficients estimated by hierarchal linear model (HLM) for DAS-28 (standard error in parentheses).

	Dependent Variable			
	DAS-28			
	(M1)	(M2)	(M3)	(M4)
Gender (male)	−0.213 ***	−0.040 ***		
	(−0.268, −0.157)	(−0.064, −0.017)		
RA Disease Duration	−0.002	−0.004 ***		
	(−0.006, 0.002)	(−0.006, −0.003)		
Nationality (non-Kuwaitis)	0.272 ***	0.022		
	(0.214, 0.331)	(−0.007, 0.051)		
Governorate (Farwaniya)	0.807 ***	0.299 ***		
	(0.666, 0.947)	(0.239, 0.358)		
Governorate (Hawally)	0.852 ***	0.277 ***		
	(0.704, 1.000)	(0.214, 0.341)		
Governorate (Jahra)	1.143 ***	0.254 ***		
	(0.933, 1.352)	(0.150, 0.359)		
Governorate (Kuwait City)	0.744 ***	0.290 ***		
	(0.606, 0.882)	(0.232, 0.348)		
Governorate (Mubarak Alkabeer)	0.955 ***	0.175 ***		
	(0.793, 1.117)	(0.106, 0.244)		
Comorbidity (Yes)	0.060 **	−0.051 ***		
	(0.007, 0.114)	(−0.074, −0.029)		
Treatment Class (cDMARDs)		0.064 ***		
		(0.036, 0.092)		
Swollen		0.090 ***		0.226 ***
		(0.085, 0.095)		(0.208, 0.244)
Tender		0.099 ***		
		(0.096, 0.101)		
RF (Positive)		0.035 ***		0.004
		(0.010, 0.060)		(−0.078, 0.085)
ACPA (Positive)		0.008		0.007
		(−0.015, 0.031)		(−0.063, 0.078)
Patient Global Assessment		0.097 ***		
		(0.088, 0.105)		
Physician Global Assessment		0.014 **		
		(0.002, 0.025)		
ESR		0.028 ***		0.035 ***
		(0.027, 0.028)		(0.034, 0.037)
CRP		0.017 ***		0.001
		(0.015, 0.020)		(−0.006, 0.009)
NO${}_{2}$			0.003 **	0.003 ***
			(0.001, 0.005)	(0.002, 0.005)
O${}_{3}$			0.002	0.003
			(−0.002, 0.006)	(−0.001, 0.006)
SO${}_{2}$			0.004 ***	0.003 **
			(0.001, 0.007)	(0.0004, 0.005)
CO			−0.051	−0.001
			(−0.114, 0.012)	(−0.053, 0.052)
PM${}_{10}$			0.0002	0.00003
			(−0.0002, 0.001)	(−0.0003, 0.0004)
Constant	1.845 ***	1.029 ***	2.586 ***	1.506 ***
	(1.701, 1.988)	(0.966, 1.093)	(2.435, 2.738)	(1.358, 1.654)
R${}^{2}$	0.034	0.865	0.007	0.488
Adjusted R${}^{2}$	0.033	0.865	0.006	0.486

Note: ** *p* < 0.05; *** *p* < 0.01.

**Table 6 ijerph-17-00416-t006:** Coefficients estimated through HLM for CDAI (standard error in parentheses).

	Dependent Variable			
	CDAI			
	(M1)	(M2)	(M3)	(M4)
Gender (male)	−0.748 ***	−0.078		
	(−1.187, −0.309)	(−0.218, 0.061)		
RA Disease Duration	−0.052 ***	0.005		
	(−0.082, −0.021)	(−0.005, 0.015)		
Nationality (non-Kuwaitis)	1.209 ***	0.255 ***		
	(0.747, 1.671)	(0.081, 0.429)		
Governorate (Farwaniya)	0.066	−1.199 ***		
	(−1.044, 1.176)	(−1.550, −0.848)		
Governorate (Hawally)	3.408 ***	0.526 ***		
	(2.241, 4.576)	(0.152, 0.900)		
Governorate (Jahra)	4.662 ***	−0.559 *		
	(3.008, 6.315)	(−1.179, 0.062)		
Governorate (Kuwait City)	1.947 ***	−0.135		
	(0.858, 3.036)	(−0.476, 0.207)		
Governorate (Mubarak Alkabeer)	4.430 ***	−0.064		
	(3.150, 5.709)	(−0.474, 0.345)		
Comorbidity (Yes)	0.984 ***	0.147 **		
	(0.564, 1.405)	(0.014, 0.281)		
Treatment Class (cDMARDs)		−0.059		
		(−0.225, 0.107)		
Swollen		1.145 ***		3.035 ***
		(1.114, 1.175)		(2.848, 3.222)
Tender		1.423 ***		
		(1.410, 1.435)		
RF (Positive)		0.025		−0.066
		(−0.123, 0.173)		(−0.901, 0.770)
ACPA (Positive)		0.226 ***		−0.594
		(0.090, 0.363)		(−1.320, 0.132)
Patient Global Assessment		1.067 ***		
		(1.059, 1.075)		
Physician Global Assessment		0.871 ***		
		(0.861, 0.881)		
ESR		0.019 ***		0.085 ***
		(0.016, 0.022)		(0.068, 0.103)
CRP		−0.050 ***		−0.171 ***
		(−0.064, −0.037)		(−0.246, −0.095)
NO${}_{2}$			0.040 ***	0.048 ***
			(0.022, 0.058)	(0.030, 0.066)
O${}_{3}$			0.027	0.039 **
			(−0.008, 0.062)	(0.003, 0.074)
SO${}_{2}$			0.044 ***	0.044 ***
			(0.018, 0.070)	(0.018, 0.070)
CO			−0.062	0.185
			(−0.601, 0.476)	(−0.358, 0.729)
PM${}_{10}$			0.001	0.0004
			(−0.003, 0.004)	(−0.003, 0.004)
Constant	4.537 ***	1.322 ***	5.306 ***	2.540 ***
	(3.404, 5.671)	(0.948, 1.697)	(4.006, 6.606)	(1.017, 4.064)
R${}^{2}$	0.030	0.925	0.015	0.299
Adjusted R${}^{2}$	0.029	0.925	0.014	0.297

Note: * *p* < 0.1; ** *p* < 0.05; *** *p* < 0.01.

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
