# Peer review of "Influence of Ambient Air Pollution on Rheumatoid Arthritis Disease Activity Score Index"

_ijerph, 2020, doi:10.3390/ijerph17020416_

Round 1
Reviewer 1 Report
Comment: The revised manuscript was reviewed and I find there are extensive changes in the format of the manuscript. There are, however, several critical additional points to be discussed and need revision as below.
There are still some points that need to be revised in the format of the manuscript.
- The introduction is not well formulated. The fifth paragraph beginning from “Moreover” should be located in the method section. The use of bullet points is not recommended and please change the format in a full sentence.
- The last sentence of the first paragraph is duplicated with the last sentence of the introduction.
- Numbers should not be prefixed in the main section titles of Introduction, Materials and Methods, Results, Discussions. The result section is not separated.
Materials and Methods- Please include a scoring method of DAS-23 and CDAI in a separate section in the Methods.
There are many critical points to discuss on the statistical methods. Analysis should be highly focused on the extracting association between air quality and RA activities. However, in spite of the bulky presentation of descriptive features, association analysis is not clear enough.- Models 1 to 4 demonstrated that the main determinant of rheumatoid arthritis (RA) activities is biologic risk factors of RA, with R square of 86.5% and 92.5%, respectively. The effect of air pollutants is relatively minor as is common to other health effects. Therefore, to demonstrate the effect of air pollutants, adjustment for the main determinants of the RA should be performed and presented. Tables 6 and 7 may be reorganized to present only the result of model 4 after controlling for variables in model 2.
- In the building model, it should consider nonlinearity of the relationship between air pollutants and health outcomes such as RA activities. Either GLM with a spline or GAM model should be considered. In addition, there should be cases with repeated measurement of RA disease activities. A mixed-effect model such as GLMM with spline should be applied if there are repeated measurement cases.
- There is no consideration of the lag effect in the model. What is the rationale of the effect of air pollution on RA activities? Depending on the assumption of wither short-term of intermediate-term effect, the lag structure should be set in days or weeks.
- Other environmental variables including temperature may be adjusted in the model.
Tables are still bulky in presenting descriptive features. Tables 2 to 4 may be merged into one or two tables presenting only descriptive features of demographics, risk factors, and air pollution exposure variables in all districts. Stratified tables presenting six districts may be presented as supplement tables or excluded. In the discussion section, please include a description of the strengths and limitations of this study. Some description of conclusions does not match with the result of this study. Where did 7% come from?
Minor points:
The citation of references in the text should be revised. e.g., “According to [18], … to any human need [19].” should be revised as “According to Johnson et al (2010), …to any human need [18.19].Author Response
Dear Reviewer 1
First of all, I would like to thank you for your review. I appreciate your time and effort for helping me with your great comments to develop my article. We really considered all your great comments seriously to change and develop the article. I have checked your great feedback and we really got great benefit from what you mentioned to us. I did the required changes according to what you have suggest, Kindly find the attachment.
Regards,
Authors

Reviewer 2 Report
Manuscript ID: ijerph-633002
I appreciate authors’ effort addressing my previous comments. The revision has improved a lot. However, I still have some major comments.
Abstract: The abstract should start with background, followed by method (data collection and analysis), results and then conclusion. I’m still confused when I’m reading this revised abstract.
In general, authors should indicate why they conducted this study in the first few sentences of abstract. For example, “Air pollution has been linked with adverse health outcomes. However, little evidence is available on its effect on Rheumatoid Arthritis Disease. “ OR, “air pollution has been related with RA, however, little evidence is available in Kuwait.”
This will help readers to understand the significance of your study.
“The significant association between the study variables included two sets of data and was determined…”
The significance of association should be based on your results. What do you mean by “significant association” here?
“ Descriptive statistics, correlation analysis, and linear regression techniques were employed to estimate the significant relationships between RA disease activity DAS-28, CDAI, and air pollution components…”
You cannot say “estimate significant relationships”. You can only suggest significance after your estimation.
“multiple imputations on the data output were also applied by the chained equations (MICE) algorithm to identify missing air pollution data. Based on their demographic information and K-EPA information database, a total of 9875 patients who live in Kuwait City and had been exposed to atmospheric pollution were also involved in the study.”
To my understanding, MI here is used for preparing air pollution data for your main analysis. So it should be put after “Moreover, air pollution data were obtained from The Kuwait Environmental Public Authority (K-EPA) during that period.”
“a total of 9875 patients who live in Kuwait City and had been exposed to atmospheric pollution were also involved in the study.”
What do you mean “also” here? Except from the patients data you mention in the 3rd sentence of abstract, did you ALSO collect other health data?
When reporting your results in abstract, it would be good to report your estimates with confidence interval (but not in conclusion part). What is your estimate for association between PM10 and disease activity? What is your estimate for association between NO2 and disease activity? How about for SO2, O3, and CO? And pls include at least 95%CI when reporting your estimates. Conclusion in abstract is more like results. Your conclusion should be interpreted from your results. E.g., The findings suggest air pollution may be a risk factor for RA.
Introduction
2nd paragraph: To be clear enough, authors should just say there is evidence relating air pollution with RA, instead of saying “human health”, since ref[3-5] all studied air pollution and RA.
My another question here is what is the difference of your current study compared with previous studies. Since there are already lots of studies on this topic, what is the strength of current study? I reckon the limited investigation in Kuwait is one of the main strengths??? Or is it your study used other indicators for RA?
Statistical analysis
The main objective of the study is to investigate the association between air pollution and RA. Then the relationship between demographic variables and RA was examined in M1. Since the association was statistically significant, then these factors should be controlled in final model or the final investigation on air pollution-RA associations should be investigated in different governorate, for example. So what is the purpose for M1 and M2?
Results
Again the main purpose of this study is association between Air pollution and RA. So in result section, report descriptive summary (including distribution) of exposure and health outcomes. And then report for association estimates. So “4. Distribution analysis for pollutants” should be move to before 6th paragraph of 3. Statistical Results.
Table 6 and 7: adjusted R2: what did you adjusted for?
Discussion:
In the first paragraph of discussion, you should summarize the key findings of your study.
Authors should compare their results with previous studies. The authors tried to compare however the discussion has to be improved. Authors should discuss why there is inconsistence between this current study and other studies, instead of listing the results from previous studies.
The strengths and limitations should be discussed as well.

Author Response
Dear Reviewer 2
First of all, I would like to thank you for your review. I appreciate your time and effort for helping me with your great comments to develop my article. We really considered all your great comments seriously to change and develop the article. I have checked your great feedback and we really got great benefit from what you mentioned to us. I did the required changes according to what you have suggest, Kindly find the attachment.
Regards,
Authors

Reviewer 3 Report
I do appreciate the answers of the authors to my raised points.
However I must still say that the raised points are not treated seriously enough. The reader can not understand the rationale of the study and the presentation of the results is not sound.
Author Response
Dear Reviewer 3
First of all, I would like to thank you for your review. I appreciate your time and effort for helping me with your great comments to develop my article. We really considered all your great comments seriously to change and develop the article. I have checked your great feedback and we really got great benefit from what you mentioned to us. I did the required changes according to what you have suggest, Kindly find the attachment.
Regards,
Authors

Round 2
Reviewer 1 Report
I find that most of the points were taken and properly revised and responded. Two additional minor points are suggested.
Introduction, P2 L2-3: This sentence does not seem to be a complete one and needs to be revised or replaced. Mention of not presenting lag effect may be included int he limitation section of the discussion.
Author Response
Dear Reviewer 1
First of all, I would like to thank you for your review. I appreciate your time and effort for helping me with your great comments to develop my article. We really considered all your great comments seriously to change and develop the article. I have checked your great feedback and we really got great benefit from what you mentioned to us. I did the required changes according to what you have suggest:
Point #1
Introduction, P2 L2-3: This sentence does not seem to be a complete one and needs to be revised or replaced.
Response #1
I remove it from introduction, because we found the sentence was not related to the main objective of the study.
Point #2
Mention of not presenting lag effect may be included int he limitation section of the discussion.
Response #2
This is updated in the limitation paragraph:
One of the Limitations in the study is related to the records of the follow up visits. As the data is extracted from a registry, the number of the hospital visits is not equal among the patients. It ranges from one to 49 visits. And because of this limitation, time series analysis during patients visits could not be performed to estimate lag effect between air pollution and RA disease activity. Single lag day effect or moving average of several previous days lag effect can not be investigated in this study because of the data layout and the study duration is very short (from 2013 to 2017). But it can be developed by improving time series features in the future.

Reviewer 2 Report
Abstract
Results: (1) You don’t need to report correlation results if you have results from regression models. (2) When reporting results from regression models, you don’t report beta directly. You should interpret the beta. For how much increase in which pollutant (e.g., 1 ug/m3 increase in daily concentratioin of PM2.5), XX will increase XXXX (95%CI:XX-XX).
Conclusion: “Conclusively, some of the important air pollution components were significantly associated with RA disease activity” This is result, not conclusion.
Pls try avoid “significant correlation/association”. You may say “ there is a strong evidence….” OR “We observed a strong association…” OR “weak evidence”.
Introduction
Pls check the order of your reference. In your introduction, your reference start from [1], but followed by [44,45].
“Point #12 another question here is what is the difference of your current study compared with previous studies. Since there are already lots of studies on this topic, what is the strength of current study? I reckon the limited investigation in Kuwait is one of the main strengths??? Or is it your study used other indicators for RA?
Response #12: We use different indicators DAS28 and CDAI, in RA, most of their studies they depend on DAS28 and CDAI in diagnosing disease activity for patient in most of rheumatoid previous studies. ”
Did you incorporate this into revised introduction?
Method: “Point #13 The main objective of the study is to investigate the association between air pollution and RA. Then the relationship between demographic variables and RA was examined in M1. Since the association was statistically significant, then these factors should be controlled in final model or the final investigation on air pollution-RA associations should be investigated in different governorate, for example. So what is the purpose for M1 and M2?
Response #13: The four model reflect:
Model 1: We are trying to estimate the demographics or patients characteristic effects on disease activity (DAS28 or CDAI).
Model 2: We are trying to estimate the demographics or patients characteristic plus rheumatoid factors (Treatment Class (cDMARDs), Swollen, Tender, RF, ACPA, Patient Global Assessment, Physician Global Assessment, ESR, CRP) effects on disease activity (DAS28 or CDAI).
Model 3: We are trying to estimate the air pollution effects on disease activity (DAS28 or CDAI).
Model 4: We are trying to estimate the air pollution effects with other patients characteristics on disease activity (DAS28 or CDAI). We choose the factors in order to get the high level of R square for more model explanation. ”
Maybe I did not explain myself clearly in my previous comments.
The main objective of the study is investigating the association between air pollution and RA. So Model 3 and 4 were developed for your main objective.
So what is the original purpose for M1 and M2? These two models (M1 & M2) were not related with the objective of this study. So I don’t understand why you developed M1 and M2. Did you adjusted gender, disease duration, nationality, governorate, comorbidity, treatment class, swollen, tender, etc. in M3 and M4??
Before regression models, did the authors check the distribution of health indictors (e.g., DAS)? Is it normal distribution? Is there any outlier? For this kind of health indicators, sometime outlier can be quite common. Did you remove the outliers in your regression models?
In your results, Paragraph 7 mentioned the normal distribution. I assume authors should have checked the distribution of response variables as well?
I think authors adjusted for other factors in M3 and M4. Pls make it clear in the method part.
Discussion
Limitations: I think the lag effect of air pollution can be considered here. Single lag day effect or moving average of several previous days’ lag effect can be investigated. But it can be investigated in the future.

Author Response
Dear Reviewer 2
First of all, I would like to thank you for your review. I appreciate your time and effort for helping me with your great comments to develop my article. We really considered all your great comments seriously to change and develop the article. I have checked your great feedback and we really got great benefit from what you mentioned to us. I did the required changes according to what you have suggest:
Point #1
(1) You don’t need to report correlation results if you have results from regression models.
Response #1
I remove it from abstract, because it has the same indication of the regression models results.
Point #2
(2) When reporting results from regression models, you don’t report beta directly. You should interpret the beta. For how much increase in which pollutant (e.g., 1 ug/m3 increase in daily concentratioin of PM2.5), XX will increase XXXX (95%CI:XX-XX).
Response #2
We update the results section as:
(e.g., 1 ug/m3 increase in daily concentratioin of NO2, DAS-28 index will increase by 0.002 (95%CI:0.002–0.005); and CDAI index will increase by 0.048 (95% CI: 0.030–0.066))
whereas for SO2, the results showed a positive significant effect with β = 0.003 (95% CI: 0.0004–0.005) and β = 0.044 (95% CI: 0.018–0.070) for DAS-28 and CDAI, respectively (e.g., 1 ug/m3 increase in daily concentratioin of SO2, DAS-28 index will increase by 0.003 with 95%CI:0.0004–0.005; and CDAI index will increase by 0.044 with 95% CI: 0.018–0.070).
Point #3
Pls try avoid “significant correlation/association”. You may say “ there is a strong evidence….” OR “We observed a strong association…” OR “weak evidence”.
Response #3
The sentence updated to:
Conclusively, we observed a strong association between air pollution with RA disease activity. This study suggests air pollution as a risk factor for RA and recommends further measures to be taken by the authorities to control this health problem.
Note: we sorted as strong because the relation or the association was confirmed under the 99% confidence level of alpha (p-value less than 0.01).
Point #4
Pls check the order of your reference. In your introduction, your reference start from [1], but followed by [44,45].
Response #4
Fixed and updated
Point #5
We use different indicators DAS28 and CDAI, in RA, most of their studies they depend on DAS28 and CDAI in diagnosing disease activity for patient in most of rheumatoid previous studies. ”
Did you incorporate this into revised introduction?
Response #5
Yes, it was mentioned in the introduction section:
Many composite indices for RA progress measurement are actually available: for example, the Disease Activity Score (DAS) with 28 examined joints (DAS-28) is one common RA index that has been extensively employed to identify the disease progress level for RA patients [13–15]. The Disease Activity Score was developed to measure and assess RA disease activity in daily clinical practice, clinical trials, and long-term observational studies [16]. The second RA index is the Clinical Disease Activity index (CDAI) [17,18], which is used to assess disease activity. The CDAI was developed to provide physicians and patients with simple and more understandable instruments.
and in section 2.2 as well:
2.2. Calculating RA indices
RA disease activity scores are measured using two different indices which are DAS-28 and CDAI. The DAS-28 is the sum of four outcome parameters, TJC28: The number of tender joints (0–28); SJC28: The number of swollen joints (0–28); ESR: The erythrocyte sedimentation rate (in mm/h); CRP: C-reactive protein (CRP) may be used as an alternative to ESR in the calculation; and GH: The patient global health assessment (from 0 = best to 100 = worst) (see eq. 1).
The second index is The Clinical Disease Activity Index (CDAI). CDAI takes into account the following items, TJC28: The number of tender joints (0–28); SJC28: The number of swollen joints (0–28); PaGH: The patient global health assessment (from 0 = best to 10 = worst); and PrGH: The care provider global health assessment (from 0 = best to 10 = worst) (see eq. 2).
Point #6
The main objective of the study is investigating the association between air pollution and RA. So Model 3 and 4 were developed for your main objective.
Response #6
we tried to explain it in the paper in the method section, we updated the method section (statistical analysis) as follow:
Regression analyses were performed separately for DAS-28 and CDAI as response variables. Four regression models were estimated to highlight the most significant variables associated with the response variables (DAS-28 or CDAI). And because of the variables are belong to two different databases, we start with the first regression (M1) that measuring the association between patients demographics (e.g. gender, RA disease duration, nationality, governorate and comorbidity) with the response variables (DAS-28 or CDAI), then we estimated the second regression model (M2) that measuring the association between rheumatoid factors after adjusting for gender, RA disease duration, nationality, governorate and comorbidity with the response variables (DAS-28 or CDAI). Model 1 and Model 2 are estimated from KRRD database. Then, we estimate model 3 and model 4 to highlight if there is any association between air pollution with disease activity indices (DAS-28 or CDAI) after merging the EPA with KRRD databases. Model 3 (M3) is measuring the direct effect from air pollutants to disease activity indices (DAS-28 or CDAI), where model 4 (M4) is measuring the association between air pollutants to disease activity indices after adjusted for rheumatoid factors that are mentioned in model 2 (M2) (e.g. comorbidity, treatment class, swollen, tender, etc.). For better data fit, model comparison techniques using deviance scores were implemented to confirm the best choice of model [34,35].
The reason why we estimated model 1 and model 2 was because of highlighting the association between rheumatoid factors with response variables (DAS-28 and CDAI) from KRRD database only. Model 3 and model 4 were examined the associations between air pollutants with disease activity indices after adjusting for other factors related to rheumatoid factors (in M2) after merging the two databases (KRRD and EPA).
Point #7
Before regression models, did the authors check the distribution of health indictors (e.g., DAS)? Is it normal distribution? Is there any outlier? For this kind of health indicators, sometime outlier can be quite common. Did you remove the outliers in your regression models?
Response #7
1- we have checked the distribution for the scale variables.
2- all outliers were removed from the model.
3- the air pollutants variables were transformed using log function for better distribution performance.
Point #7
In your results, Paragraph 7 mentioned the normal distribution. I assume authors should have checked the distribution of response variables as well?
Response #7
yes, this is true, we checked for response variables and we found that the distribution of the disease activities are both follow normal distribution using QQ plot.
Moreover, all pollutants distribution are positively skewed (e.g. mean are higher the median for all pollutants). However, log transformation was employed to all pollutants in the regression model for better and quality parameter estimation. Logarithmic transformation pulls extreme values in the pollutants into the normal distribution. In the regression modeling, since the logarithmic transformation is not considered as linear explanation of the regression model is made easier to transform the other all variables in the model with the same logarithmic transformation. However, this is not necessary as long as all other variables achieved the normality assumption.
Point #8
Discussion ==> Limitations: I think the lag effect of air pollution can be considered here. Single lag day effect or moving average of several previous days’ lag effect can be investigated. But it can be investigated in the future.
Response #8
This is updated in the limitation paragraph:
One of the Limitations in the study is related to the records of the follow up visits. As the data is extracted from a registry, the number of the hospital visits is not equal among the patients. It ranges from one to 49 visits. And because of this limitation, time series analysis during patients visits could not be performed to estimate lag effect between air pollution and RA disease activity. Single lag day effect or moving average of several previous days lag effect can be investigated. But it can be investigated in the future.

Reviewer 3 Report
Major concerns of the reviewer have been explained by the authors.
It remains a critical point that the time frame of the study is extremely short (2013-2017). Additionally the total of 1651 RA patients in this short time frame is also a shortcoming of the study.
Author Response
Dear Reviewer 3
First of all, I would like to thank you for your review. I appreciate your time and effort for helping me with your great comments to develop my article. We really considered all your great comments seriously to change and develop the article. I have checked your great feedback and we really got great benefit from what you mentioned to us. I did the required changes according to what you have suggest:
Point #1
It remains a critical point that the time frame of the study is extremely short (2013-2017). Additionally the total of 1651 RA patients in this short time frame is also a shortcoming of the study.
Response #1
Yes, that is true we agreed that the period is short and we have 1651 RA patients but with more than 7000 individual visits. Also, we mentioned the limitation that our database layout can not measures the lag effect between air pollution and RA disease activity; and we will develop the database to be more efficient with time series features.
the sentence was developed as follow:
One of the Limitations in the study is related to the records of the follow up visits. As the data is extracted from a registry, the number of the hospital visits is not equal among the patients. It ranges from one to 49 visits. And because of this limitation, time series analysis during patients visits could not be performed to estimate lag effect between air pollution and RA disease activity. Single lag day effect or moving average of several previous days lag effect can not be investigated in this study because of the data layout and the study duration is very short (from 2013 to 2017). But it can be developed by improving time series features in the future.

This manuscript is a resubmission of an earlier submission. The following is a list of the peer review reports and author responses from that submission.
Round 1
Reviewer 1 Report
This paper presents an interesting finding on the relationship between rheumatoid arthritis disease activity represented by DAS28 and air pollution in Kuwait. In spite of its theme and interesting result, I would not agree to consider the publication of this paper for the following reasons. I recommend resubmission of the manuscript after fundamental revision.
The hypothesis of the authors on the association between air pollutant(s) and RA activity is not clear. The authors did not clarify whether air pollutants are risk factors on the development of RA or triggering factors in RA patients. The study design is fitted for the latter, but in the text, they are many confusion or a vague description of this relationship. It is not clear whether SO2 is a problem. In the abstract and result of the main text, the authors are mainly presenting NO2 and other pollutants. The title is misleading and does not represent the result of the paper. The manuscript does not follow the format of the IJERPH, which is common for the standard format of the scientific paper in the health field. The introduction is too long and long list of a literature review is not well focused. Discussion is too short and discussions about the methodology and result of this study are either sketchy or missing. The abstract should be written in one paragraph if it is unformatted. There are many typos and abuse of upper case letters. Some of the documents were not properly cited. Some of the citations are either gray literature or non-peer reviewed opinion (L31-33). There is confusion in the use of terminology (e.g. rheumatoid arthritis disease vs. rheumatoid arthritis or rheumatic diseases, L34). Some of the incomplete sentences are present (L101). Inappropriate citation format (L108) The description of the source of data is too sketchy and does not present the details of data linkage between air pollution and RA visits data. There is no mention of ethical clearance. The description of the statistical method is highly detailed, but intermediate results should be omitted. Description of the key findings of this study is too sketchy.Reviewer 2 Report
This study aims to study the health impact of air pollution on RA. The author did a lot of analysis. However, this manuscript was not well written and the results are lack of discussion.
Abstract:
Pls first describe the background of this study.
The abbreviation for Particulate Matter is not PM10. Pls provide full term for PM10.
Pls provide the full term for RA and DAS when it first appears in your abstract.
The method and results in your abstract was not clearly described. It is difficult to have a general idea of your study and get the key information from your abstract.
Please revise the abstract carefully.
The conclusion in your abstract is also not clear. Your aim of this study as you stated in your first sentence is to investigate the effect of air pollution on RA. So it is difficult for me to link your aim with the 2nd sentence in conclusion.
Introduction:
Not well organized. You should clearly pointed out the knowledge gaps in your current topic and relate it to the significance of your study.
Methodology:
4.1. Data collection
Air pollutants data were collected from how many stations? Where are these stations? Close to the hospitals? And did you collect hourly data? or daily data? And especially for O3, is it maximum 8hour data?
4.2
You don’t need to include basic background knowledge in your main text. And also you should include reference when you are explaining these background knowledge.
Did you consider lag effect of air pollutants here? and explain your reasons.
If you used Multiple Imputation, you should at least mention it in your abstract.
You did not explain clearly why you use multiple imputation to deal with missing values.
For missing values, you have to know how many missing data, and the distribution of missing data, and also if they are missing randomly or not, and more importantly, the reasons why they are missing. These are all important points you need to consider before you do anything with your missing data.
And probably you need a sensitivity analysis to compare your main results before and after your treatment with missing data.
Results
The results should start with descriptive analysis of your data, including description on how many data are missing in each variable. For the results on multiple imputation, you don’t need to put everything here. you can describe briefly and leave most of the information in the appendix. Otherwise, the paper will be too long ,and readers will get confused.
For table 2 and 3, if you merged health data from all six hospitals together, then you can report descriptive results for whole study participants. Leave the detailed information in appendix.
Move Table 5 to appendix
Your main aim of this study is to investigate the relationship between air pollution and RA. However, you listed way too many results which are supposed to be included in appendix. With all these information, your main objective of this study is blurred.
It’s difficult to follow.
Conclusion:
You should have a separate section for discussion.
Conclusion section is supposed to summarize the key outcomes of the study and its implications concisely.
You should point out the limitations of your study as well.
In whole paper, sometimes you use full term and sometimes you use abbreviations. Pls provide the full term and its abbreviation when the word first appears.
Reviewer 3 Report
This is an interesting research approach to RA and exposure to SO2 and PM10.
However there serious objections;
a) The rationale of the study is not clear
b) The time slot of four years (2013-2017) is not sound
c) There is a big mess and confusion about the findings worldwide compared to Kuweit
d) The discussion and conclusion chapter are not sound